# Genome-Wide Identification and Evolutionary and Mutational Analysis of the *Bos taurus Pax* Gene Family

**DOI:** 10.3390/genes15070897

**Published:** 2024-07-09

**Authors:** Jintao Zhong, Wenliang Wang, Yifei Li, Jia Wei, Shuangshuang Cui, Ning Song, Yunhai Zhang, Hongyu Liu

**Affiliations:** 1College of Animal Science and Technology, Anhui Agricultural University, Hefei 230036, China; zhongjintao2384@gmail.com (J.Z.); wwl199811231877@gmail.com (W.W.); liyifei19937207587@163.com (Y.L.); wj142028@163.com (J.W.); 1074375597@stu.ahau.edu.cn (S.C.); songning@ahau.edu.cn (N.S.); yunhaizhang@ahau.edu.cn (Y.Z.); 2Anhui Provincial Laboratory of Local Livestock and Poultry Genetical Resource Conservation and Breeding, Anhui Agricultural University, Hefei 230036, China

**Keywords:** *Bos taurus*, *Pax*, evolution, gene structure, motif

## Abstract

*Bos taurus* is known for its tolerance of coarse grains, adaptability, high temperature, humidity, and disease resistance. Primarily, cattle are raised for their meat and milk, and pinpointing genes associated with traits relevant to meat production can enhance their overall productivity. The aim of this study was to identify the genome, analyze the evolution, and explore the function of the *Pax* gene family in *B. taurus* to provide a new molecular target for breeding in meat-quality-trait cattle. In this study, 44 *Pax* genes were identified from the genome database of five species using bioinformatics technology, indicating that the genetic relationships of bovids were similar. The *Pax3* and *Pax7* protein sequences of the five animals were highly consistent. In general, the *Pax* gene of the buffalo corresponds to the domestic cattle. In summary, there are differences in affinity between the *Pax* family genes of buffalo and domestic cattle in the *Pax1/9*, *Pax2/5/8*, *Pax3/7,* and *Pax4/6* subfamilies. We believe that *Pax1/9* has an effect on the growth traits of buffalo and domestic cattle. The *Pax3/7* gene is conserved in the evolution of buffalo and domestic animals and may be a key gene regulating the growth of B. taurus. The *Pax2/5/8* subfamily affects coat color, reproductive performance, and milk production performance in cattle. The *Pax4/6* subfamily had an effect on the milk fat percentage of B. taurus. The results provide a theoretical basis for understanding the evolutionary, structural, and functional characteristics of the *Pax* family members of *B. taurus* and for molecular genetics and the breeding of meat-production *B. taurus* species.

## 1. Introduction

Domestic cattle belong to the subfamily Bovidae of the even-toed ungulaceae and were one of the earliest-domesticated domestic animals in the course of evolution. The consensus on the origin of domestic cattle is that Bos primigenius, which was widespread in Eurasia in prehistoric times, is their ancestor. Bos primigenius is divided into three continental subspecies, B.p. primigenius, B.p. Nomadicus, and B.p. opisthonomus, according to the shapes and sizes of their horns. The family consists of two species, the Taurine and the Zebuine. Up to now, a large number of studies have shown that the common Auroprotozoa and the tuberoprotozoa have undergone multiple independent domestication events. The ancestors of Zebu cattle were domesticated in the Indus Valley about 8000 years ago [1]. Common cattle were domesticated in Southwest Asia (present-day Turkey) about 10,000 years ago. The earliest archaeological evidence for the domestication of auropodial cattle is 10,750–10,250 years old [2]. In recent years, the availability of high-quality reference genomes and moderately dense genotyping marker sets has stimulated a series of genome-wide studies on plant diversity, evolutionary history, production traits, and functional elements. As research advances and molecular knowledge continues to be integrated into breeding programs, the global domestic cattle population will gradually improve and strengthen in terms of yield, environmental adaptability, and disease resistance [3].

With the advent of modern molecular biology techniques, paired box (Pax) proteins were identified by researchers as being able to regulate gene transcription during biological development [4]. Regarding the Drosophila pared (prd) gene, it was discovered that both the Drosophila prd and gooseberry (gsb) genes contain a conserved pared domain of 128 amino acids [5]. These groundbreaking discoveries inspired us to understand the highly conserved transcription factors in the *Pax* family. In vertebrates, the *Pax* gene family is divided into four subfamilies, *Pax1/9*, *Pax2/5/8*, *Pax3/7,* and *Pax4/6*, based on the different structural features of the genes [6]. Members of the *Pax* gene family are highly conserved throughout evolution and play critical roles in the organism, including the ability to coordinate the development of tissues and organs and to maintain cellular properties [7,8].

Members of the *Pax* gene family also play important roles in mammals, including the formation of the nervous system and the development of tissues and organs [9]. Previous studies of the *Pax* gene family have not compared different species or explored the evolutionary relationships between gene families in different species. While genomic studies of species have historically been laborious and distant, gene family analysis has become an emerging trend. As large numbers of species genomes have been measured and data released, the identification of gene family members from species genomic data using specific structural models has become increasingly common [10]. However, there is currently no comprehensive genome-wide study of the *Pax* gene family in *Bos taurus* that is being conducted by researchers. The purpose of this study was to identify the members of the *Pax* gene family in *B. taurus* and characterize their physicochemical properties. The joint analysis of *Pax* genes from related species was conducted to assess the functionality of *Pax* gene family members in *B. taurus*.

## 2. Materials and Methods

### 2.1. Identification of the Pax Gene Family

To analyze the *Pax* gene family, from the NCBI database, we downloaded the *B. taurus* (GCA_002263795.3), *Bos indicus* (GCF_000247795.1), *Bubalus bubalis* (GCF_019923935.1), *Sus scrofa* (GCA_000003025.6), and *Ovis aries* (GCA_002742125.1) genome files (fa) and annotation files (gff).

To identify potential *Pax* genes in these domestic animal species, using Pfam database (https://www.ebi.ac.uk/interpro/entry/pfam/) (assessed on 17 May 2024), a conservative HMG *Pax* structure domain protein sequence of the Hidden Markov model (Hidden Markov model, HMM) map was used. The model number was PF00292. The hmmsearch tool was used to search the domain sequence of the target species, and the E value was set to 1.2 × 10^−28^. The ClustalW tool was used to conduct multiple-sequence comparison of the results, and the hmm model was established again with the comparison results. Finally, the domain was searched again according to the newly established model. Genes with E-value < 0.01 were selected as members of *Pax* gene family. These were identified on the sequence of gene family members submitted to NCBI BatchCD–Search (https://www.ncbi.nlm.nih.gov/cdd) (assessed on 17 May 2024) for structural domain analysis and blast (https://blast.ncbi.nlm.nih.gov/Blast.cgi) (assessed on 17 May 2024), confirming *Pax* gene structure domain after complete and accurate evaluation and subsequent analysis process.

### 2.2. Multiple-Sequence Alignment and Phylogenetic Analysis of Pax Gene Family

The ClustalW function in MEGA11 v11.0.13 software was used to compare *Pax* gene family sequences of domestic cattle, Zebu cattle, buffalo, pig, and sheep with default parameters, and the phylogenetic evolutionary tree of related species was constructed by adjacency method (NJ) and bootstrap setting 1000 times. And we imported the results to iTOLv6 (https://itol.embl.de/) (assessed on 17 May 2024) to visualize the results.

### 2.3. Sequence Analysis of Pax Gene in B. taurus

After obtaining *Pax* gene family sequences, we used ClustalW in MEGA11 to compare *Pax* sequences of domestic cattle and make NJ evolutionary tree. In addition, we uploaded the domain data of the *Pax* gene family to MEME (Multiple-Expectation Maximization for Motif Elicitation) website (https://meme-suite.org/meme/) (assessed on 18 May 2024) to obtain the motif characteristics of the family. The minimum and maximum widths of the motifs were set to 6 and 50, respectively, and the number of search motifs was set to 10. The exon–intron pattern of *Pax* gene was obtained by analyzing the structure of *Pax* gene using GSDS website (http://gsds.gao-lab.org/) (assessed on 18 May 2024).

### 2.4. Physicochemical Properties and Subcellular Localization of Pax Gene Family in B. taurus

In order to explore the physicochemical properties of *Pax* gene family in domestic cattle, we uploaded *Pax* sequences to ExPasy (https://web.expasy.org/compute_pi/) (assessed on 20 May 2024) and obtained the length, molecular weight (MW), and isoelectric point (PI) data of *Pax* gene.

In order to determine the location of *Pax* gene family members in cells, we uploaded the amino acid sequences of *Pax* gene family members to the WoLF PSORT (https://wolfpsort.hgc.jp/) (assessed on 20 May 2024) website and selected animal protein species to analyze and predict the subcellular localization of *Pax* gene family members.

### 2.5. Co-Linearity Analysis and Chromosome Localization of Pax Gene in B. taurus

In order to determine the specific location of each *Pax* gene family member, the genome annotation files (gff) and gene list files of *B. taurus*, *B. indicus*, and *B. bubalis* downloaded from NCBI were inputted into MCScanX to achieve chromosomal localization of the *Pax* gene family. In order to explore the link between *Pax* gene family members of *B. taurus*, *B. indicus,* and *B. bubalis*, we imported genome files and annotation files of *B. taurus*, *B. indicus*, and *B. bubalis* into TBtools-II v2.102 and analyzed the data through One-Step MCScanX plug-in of the software. The Dual Systeny Plot for MCScanX plug-in was used to visualize the results to obtain the co-linearity maps of *Pax* genes of *B. taurus*, *B. indicus*, and *B. bubalis*.

### 2.6. Multiple-Sequence Alignment of Pax Gene in B. taurus

In order to explore the structural characteristics of the protein encoded by the *Pax* gene family in domestic cattle, MEGA11 ClustalW function was used to perform multiple-sequence alignment of Pax protein sequences of domestic cattle, and the comparison results were imported into GeneDoc v2.7 software for analysis and visualization.

### 2.7. Protein Three-Dimensional Structure and Interaction Network of Pax Gene

In order to explore the three-dimensional protein structure of *Pax* gene, we imported the amino acid sequence of *Pax* gene into SWISS-MODEL website to predict it. We selected the template with high coverage and similarity in the prediction results. We selected a template with a large GMQE value and a QMEAN close to 0. After saving, we built up 3D model file and PDB file and used SAVES (https://saves.mbi.ucla.edu/) (assessed on 22 May 2024) site in the model analysis to judge the PDB file availability of three-dimensional structure.

The protein interaction network of *Pax* gene was constructed using the online website STRING (https://cn.string-db.org/) (assessed on 22 May 2024) to explore the interactions among the proteins encoded by the *Pax* gene family and the results were imported into the software for beautification.

## 3. Results

### 3.1. Systematic Evolution of the Pax Gene Family

Using the hmmer tool, nine *Pax* genes were found in the whole-genome data of domestic cattle, the Zebu, buffalo, and sheep, and eight *Pax* family members except *Pax8* were found in pigs, with a total of 44 genes. According to the evolutionary relationships, these genes can be divided into four subfamilies, namely the *Pax3/7* subfamily, *Pax4/6* subfamily, *Pax1/9* subfamily, and *Pax2/5/8* subfamily. The selective evolutionary analysis of the five animals showed that domestic cattle were more homologous to buffalo and the Zebu than other domestic animals most of the time (Figure 1).

### 3.2. Gene Structure Characterization of Pax Gene Family in B. taurus

We analyzed the structure of the *Pax* gene family in domestic cattle. The results included the gene structure (Figure 2), phylogenetic tree (Figure 3A), motif (Figure 3B), conserved domain distribution (Figure 3C), and differentially conserved motifs in *B. taurus*’s *Pax* gene family (Table 1). Gene structure analysis found that the number of exons and introns in each *Pax* gene was fixed, *Pax9* had the least exons (four), and *Pax2* had the most exons (thirteen). The number of UTRs ranged from one to five (Figure 2). A total of 10 motifs were identified from the *Pax* gene family of domestic cattle. The motifs of the *Pax* gene family ranged from 14 to 94 in length, and motif1, motif5, and motif7 all had high amino acid quantities. Motif1 and motif2 are present in every gene family member. Through the study of motifs, we found that in the same gene subfamily, the number of motifs is often consistent. For example, the *Pax3/7* subfamily has a relatively large number of motifs while the *Pax1/9* subfamily has a relatively small number of motifs (Figure 3B). All *Pax* genes contain *Pax* domains; the homeodomain superfamily is found only in the *Pax2* gene and the AKR_SF superfamily is found only in *Pax4*. The species and quantity distributions of conserved domains of genes within the same subfamily are also very similar (Figure 3C).

### 3.3. Physical and Chemical Properties and Subcellular Localization of Pax Gene Family in B. taurus

ExPasy was used to analyze the lengths, molecular weights (MWs), and isoelectric points (PIs) of the *Pax* gene family. The results showed that the lengths of the *Pax* gene family ranged from 328 to 505 bp and the molecular weights ranged from 35,573.8 to 55,100.5 Da. The isoelectric points (PIs) ranged from 7.93 to 10.61, all of which are alkaline (Table 2).

We used the WoLF PSORT website for the subcellular localization of the *Pax* gene family members and the results showed that all members of the *Pax* gene family were located in the nucleus (Table 2).

### 3.4. Co-linearity Analysis and Chromosome Localization of Pax Gene in B. taurus

Using MCScanX, we located members of the *Pax* gene family in *B. taurus*, *B. indicus*, and *B. bubalis*. The analysis results (Figure 4) indicate a highly consistent distribution of *Pax* genes between *B. taurus* and *B. indicus*. This suggests a potentially high functional similarity between the *Pax* gene families in *B. taurus* and *B. indicus*, implying that studying the *Pax* gene family in *B. indicus* could help in assessing the functionality of the *Pax* gene family in *B. taurus*.

Co-linearity analysis provides valuable insights into evolutionary relationships and polyploid events. According to the previous evolutionary tree of the *Pax* gene family, *B. taurus*, *B. bubalis*, and *B. indicus* exhibit a closer genetic affinity. Therefore, through the co-linearity analysis of these three cattle species and considering the known functions of *Pax* genes in *B. bubalis* and *B. indicus*, we aimed to explore the functions of *B. taurus Pax* genes that exhibit co-linearity. Using TBtools, we generated graphical representations of the co-linearity analysis results for these species. *B. bubalis* shows co-linearity with *B. taurus* and *B. indicus Pax* genes, displaying an overall one-to-one correspondence across different species (Figure 5). Among them, *pax* gene family members in B. taurus and B. indicus also exist on the same chromosome, which indicates that Pax gene differentiation in B. taurus and B. indicus has not gone far. This suggests potential functional similarity between *Pax* gene family members in *B. taurus* and *Pax* gene family members in *B. indicus*. The *Pax* gene function of *B. taurus* could be evaluated based on the *Pax* gene function in *B. indicus* and *B. bubalis.*

### 3.5. Multi-Sequence Alignment of Pax Protein in B. taurus

The ClustalW function of MEGA11 was applied to compare the protein sequence of the *Pax* gene in domestic cattle and the results were imported into GeneDoc to produce the following results (Figure 6). The red area in the figure below indicates that the comparison rate is 100%, the orange area has a comparison rate of 70% to 99%, the yellow area has a comparison rate of 50% to 69%, and the colorless area is less than 50%. The comparison results show that nine *Pax* protein sequences had high consistency in the 100–220 region and were highly conserved in this region. It was speculated that the conserved domain of the *Pax* family was located in this region, which was related to the function of the *Pax* protein.

### 3.6. Three-Dimensional Structure of Pax Protein

The SWISS-MODEL website was used to predict and select the three-dimensional structure of the Pax protein, which was evaluated by the website of SAVES for usability, as shown in the figure below (Figure 7). The three-dimensional structure of the protein is composed of a motif and domain. The activity and function of proteins are not only determined by the primary structure of protein molecules but also closely related to their unique spatial structures. The wrong spatial structures in proteins can lead to reduced function or even the inactivation of proteins, which can also lead to a range of diseases. For example, mad cow disease is caused by the aggregation of certain proteins after the misfolding, forming amyloid fiber precipitation against proteolytic enzymes, resulting in toxicity and disease.

### 3.7. Protein Interaction Network of Pax Gene

The protein interaction network of *Pax* genes was constructed using the online website STRING and we imported the results into cytoscape software for beautification (Figure 8). The results showed that only the *Pax3* and *Pax7* genes had direct interaction in the *Pax* gene family and the interactions between other *Pax* genes needed to be completed through the transfer of genes outside the family.

## 4. Discussion

*B. taurus* is one of the world’s most important livestock, a cornerstone of the world’s livestock industry, providing beef as an important source of protein for humans. *B. taurus* was one of the first domesticated animals to have spread around the world with human migration and trade, allowing it to genetically adapt to different climatic conditions in different regions [3,11]. *B. taurus* provides traction for farmers, improves agricultural production efficiency, and made an important contribution to the development of farming civilization. The *Pax* gene family is closely related to the growth traits of *B. taurus* [12]. The study of the *Pax* gene family can further increase beef yield to meet the increasing beef consumption demand of people. The evolution and gene structure of the *Pax* gene family were studied to provide a theoretical basis for breeding *B. taurus* with better meat quality traits.

Our study identified 44 members of the *Pax* gene family in five species by bioinformatic techniques. In this study, a phylogenetic analysis of *Pax* genes from species was performed to explore the differences in affinities, divergences, and motifs. Phylogenetic analyses of *Pax* family genes provided an in-depth understanding of the evolution of the *Pax* family genes. Neighbor-joining tree analysis showed that *Pax* family genes could be divided into four taxa, and the affinities of *Pax* genes differed among species of different genera within each taxon. The results showed that *B. taurus* genes have diverged during the evolution of the species and that not all genes are related in the same way. The affinities of *Pax* genes varied within the *Pax1/9*, *Pax2/5/8*, *Pax3/7*, and *Pax4/6* subfamilies among buffalo and domestic cattle.

*Pax1* is essential for the differentiation of the thymus, vertebrae, and cartilage and the maturation of chondrocytes during embryonic development [13]. In African clawed toad embryos, *Pax1* is detected early in somitogenesis and expressed at increased levels in the osteogenesis and endodermal pharyngeal sacs [14]. *Pax1* homologs (*Pax1*a and *Pax1*b) in zebrafish are also expressed in the developing osteogenesis and endodermal pharyngeal sacs [15]. *Pax9* is one of the best-characterized transcription factors involved in human tooth development, capable of influencing the number, position, and morphology of an individual tooth. Mutations in the *Pax9* gene have been reported to be associated with various types of dental hypoplasia and other inherited dental defects or variants [16]. The *Pax1/9* gene in *B. taurus* is presumed to be functionally similar to the *Pax1/9* gene in buffalo and the Zebu. The expression of the paired-box genes *Pax1* and *Pax9* is associated with limb skeleton development [4]. We speculate that *Pax1/9* affects the growth traits of *B. taurus*.

*Pax2* is essential for the development of the genitourinary system, neural tube, optic vesicles, optic cup, and optic tract [17], and *Pax2* mutations could cause eye defects [18]. The *Pax2* gene has significant genetic effects on disease resistance in buffalo and the milk fatty acids of dairy cattle [19,20]. *Pax5* is a key transcription factor that determines β-cell stereotypy and development. *Pax5* represses genes inappropriate for the β-cell lineage and induces gene expression required for β-cell development. Moreover, *Pax5* post-transcriptionally downregulates the expression of phosphatase and tensin homolog (PTEN) to promote the differentiation and survival of mature β-cells, thereby promoting humoral immunity [21,22]. *Pax5* affects coat color, which might drive the differences among Chinese yellowish coated breeds and those in the greater Far East region [23,24]. *Pax8* belongs to a class of spectrum survival genes that are required for both the normal development of certain tissues and the proliferation of cancer cells [24,25]. *Pax2* and 8 are over-represented in biological processes related to kidney organogenesis in beef-cattle placental tissues [26]. The *Pax2/5/8* subfamily affects coat color and reproductive and milk performance in cattle, but there is no clear understanding of the *B. taurus* meat traits.

The transcription factors encoded by *Pax3* and *Pax7* are among the first to be expressed in the embryo, and both are key regulators of myogenesis capable of influencing the development of mammalian limb and most hypothenar muscles [27]. *Pax3*-positive muscle stem cells become sensitive to environmental stress when *Pax3* function is impaired, and the *Pax3*-mediated induction of mammalian targets of rapamycin complex 1 (mTORC1) is required for protection [28]. *Pax3* could regulate the neural crest and the onset of myogenic differentiation in the somites, both of which represent the synchronous development of the neural crest and skeletal muscle lineages [29,30]. *Pax7* plays key roles from early central nervous system development to brain maturation and drives the formation of the vertebrate visual system by determining the formation and detection of the superior colliculus [31]. *Pax3* plays an important role in early embryonic skeletal muscle formation and can regulate the myogenic-determinant myogenic factor 5 (Myf5) and myogenic differentiation (MyoD). *Pax7* plays a dominant role in adult growth and muscle regeneration, and *Pax7* deficiency results in satellite cell deficiency, muscle atrophy, and early postnatal mortality in mice [32,33]. The polymorphism of *Pax3* affects the growth traits of Chinese domestic cattle [12]. In addition, *B. taurus Pax7* is closely related to buffalo *Pax7* and Zebu *Pax7*. The *Pax3/7* gene is conserved during the evolution of B. taurus and may be a key gene in regulating bovine growth.

*Pax4* is a direct target of neurogenin 3, which regulates β-cell specificity and is a key gene in pancreatic development. Mice lacking *Pax4* develop severe hyperglycaemia and die within days of birth due to a lack of mature pancreatic cells [34]. *Pax6* is a major transcription factor in early eye development and is extremely important in the postnatal development of the eye [35]. Human *Pax6* shares 90% and 96% sequence homology with Drosophila and zebrafish, respectively. During eye development, two structurally similar *Pax6* isoforms perform two distinct functions by activating or repressing different target genes, and both the knockout and overexpression of *Pax6* inhibit normal eye development [36]. *Pax4* regulates bovine adipocyte differentiation and lipid homeostasis and *Pax6* is an important transcription factor affecting the milk fat traits of dairy cattle [37,38]. *Pax4* affects tropical adaption between Indian Zebu cattle and riverine buffalo [39].

## 5. Conclusions

In this study, nine *Pax* genes were identified from the genome of *B. taurus*, and these could be divided into four subfamilies: *Pax1/9*, *Pax2/5/8*, *Pax3/7*, and *Pax4/6*. The *Pax* gene family members were located in the nucleus. Molecular phylogenetic analysis showed that the structures and sequences of *Pax* genes in buffalo, the Zebu, and *B. taurus* were similar. The species and quantity distributions of the domains and motifs of the subfamily members were consistent. In addition, evolutionary analysis revealed that the *Pax* domain is highly conserved in all *Pax* gene family members. Protein interaction network analysis showed that only the *Pax3* and *Pax7* genes had direct interaction among the *Pax* genes. We believe that *Pax1/9* has an effect on the growth traits of buffalo and domestic cattle. The *Pax3/7* gene is conserved in the evolution of buffalo and domestic animals and may be a key gene regulating growth in B. taurus. The *Pax2/5/8* subfamily affects coat color, reproductive performance, and milk production performance in cattle. The *Pax4/6* subfamily has an effect on the milk fat percentage of B. taurus. In future studies, the role of the *Pax* gene family on *B. taurus* needs to be verified in bovine myoblasts or in vivo. This study’s results will provide guidance for further marker-assisted selection breeding tar-geting the *Pax* gene family to improve *B. taurus’* yield performance.

## Figures and Tables

**Figure 1 genes-15-00897-f001:**
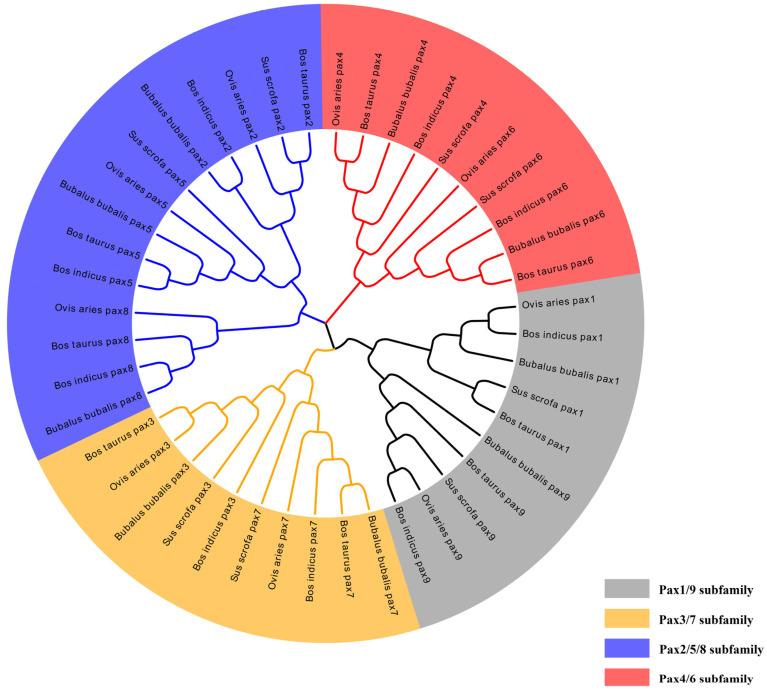
NJ evolutionary tree of *Pax* gene family in domestic cattle, Zebu, buffalo, sheep, and pig.

**Figure 2 genes-15-00897-f002:**
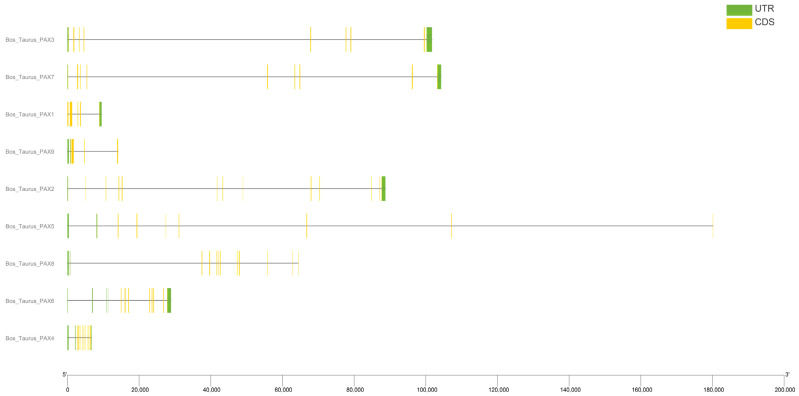
Gene structures of *B. taurus Pax* gene family.

**Figure 3 genes-15-00897-f003:**
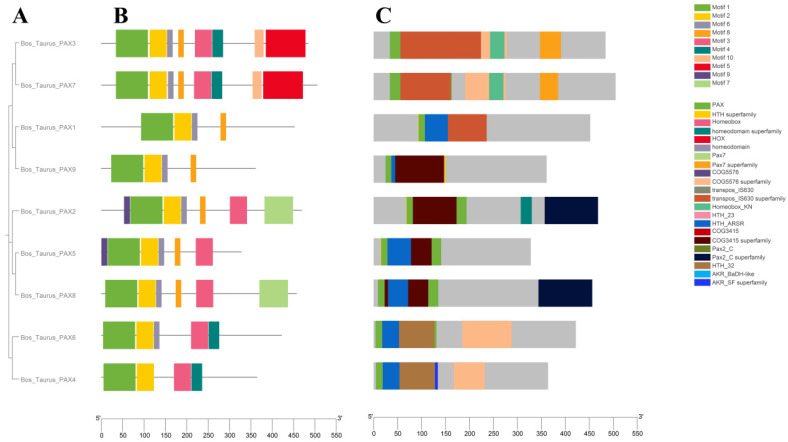
*B. taurus Pax* gene family: (**A**) phylogenetic relationship, (**B**) motif orientation, and (**C**) conserved domain distribution.

**Figure 4 genes-15-00897-f004:**
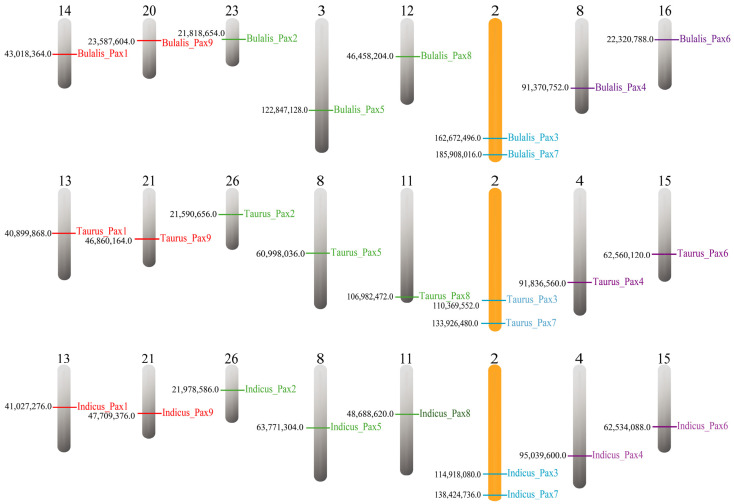
Chromosomal assignment of the *Pax* gene family.

**Figure 5 genes-15-00897-f005:**
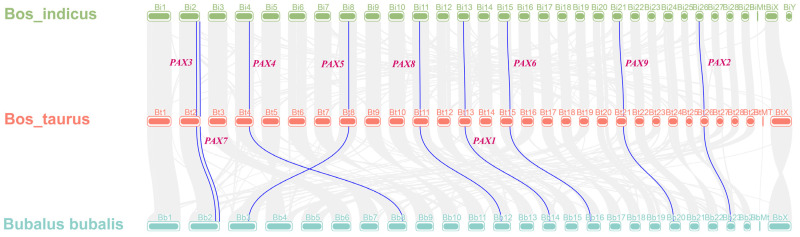
Co-linearity analysis of the *Pax* gene family in *B. taurus*, *B. indicus*, and *B. bubalis*.

**Figure 6 genes-15-00897-f006:**
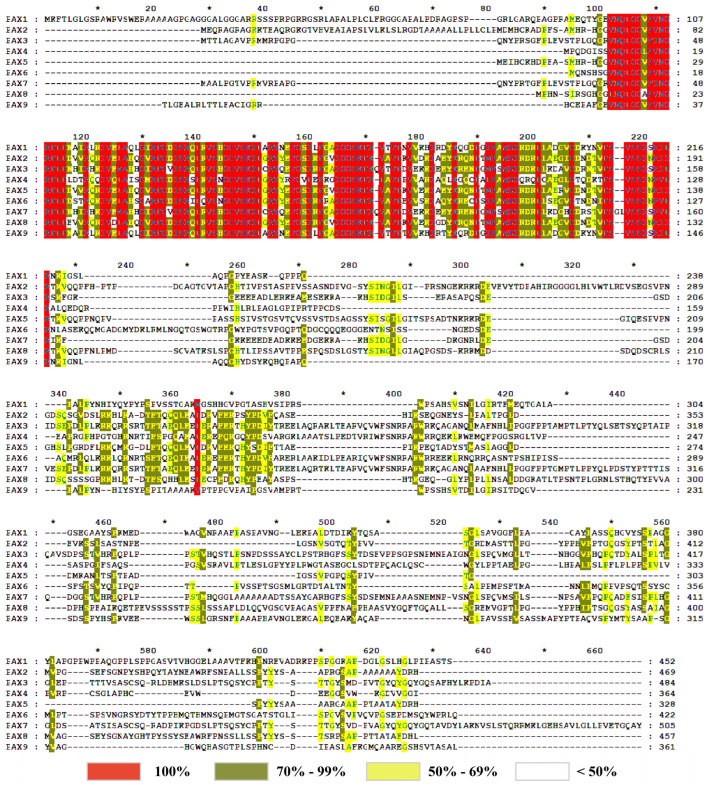
Multiple-sequence alignment of the *Pax* gene family in *B. taurus*.

**Figure 7 genes-15-00897-f007:**
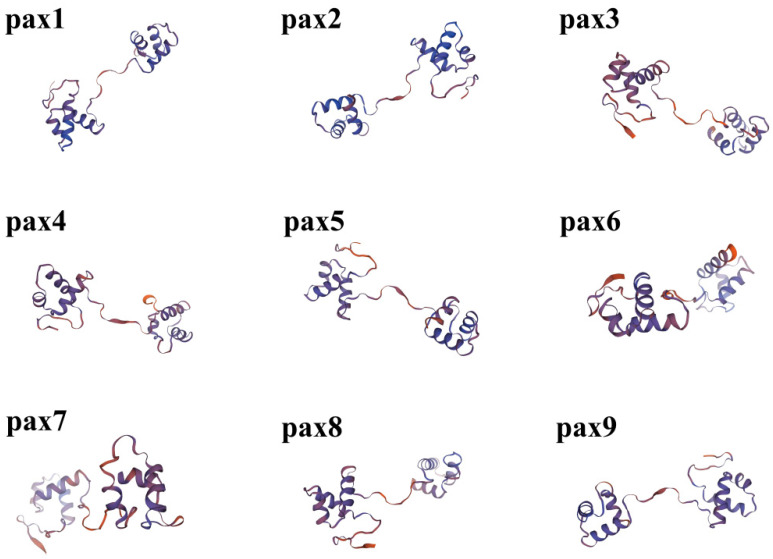
Three-dimensional structures of *Pax* gene family proteins.

**Figure 8 genes-15-00897-f008:**
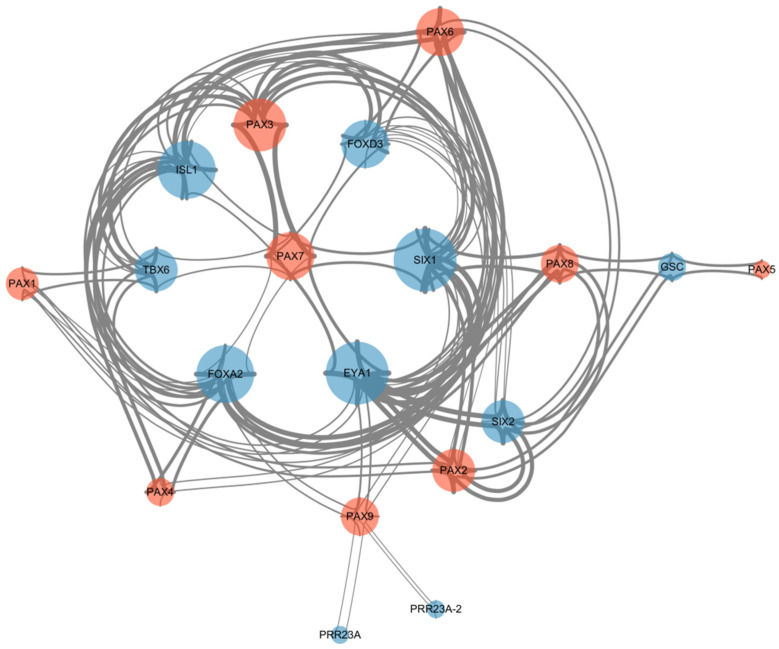
Protein interaction network of the *Pax* gene family.

**Table 1 genes-15-00897-t001:** Differentially conserved motifs in *B. taurus Pax* gene family.

Motif	Protein Sequence	Length	E-Value
MEME-1	HGGVNQLGGVFVNGRPLPBVIRQRIVELAHQGIRPCDISRQLRVSHGCVSKILGRYYETGSIRPGAIGGSKPRVAT	76	4.7 × 10^−377^
MEME-2	VVKKIAEYKRZNPGMFAWEIRDRLLAEGVCDNDTVPSVSSI	41	4.20 × 10^−182^
MEME-3	HRRRTTFTQZQLEALEKEFERTHYPDIYTREELAKREQLPE	41	4.70 × 10^−68^
MEME-4	ARVQVWFSNRRAKWRKQEGLNQLMAF	26	8.40 × 10^−27^
MEME-5	NGLSPQVMGJLSNPGGVPPQPQADFAJSPLHGGLEPATSISASCSQRADPIKPGDSLPTSQSYCPPTYSTTGYSMDPVAGYQYGQYGQSAFDYL	94	1.50 × 10^−13^
MEME-6	NRIJRTKVGQPEZQ	14	5.80 × 10^−11^
MEME-7	REMVGPTLPGYPPHIPPSGQGSYPSSAJAGMVPGSEFSGNPYGHPPYSAYNEAWRFPNPALLSSPYYY	68	7.70 × 10^−9^
MEME-8	SKPSSHSINGILGI	14	3.30 × 10^−6^
MEME-9	MEIHCKADPFAAMHR	15	6.90 × 10^−6^
MEME-10	RHGFSSYSDSFMNPAGPSNPMN	22	2.90 × 10^−2^

**Table 2 genes-15-00897-t002:** Physicochemical properties and subcellular localization of the *Pax* gene family in *B. taurus*.

No.	Gene Name	Chr	Length	MW (Da)	pI	PSL
1	B. taurus_*Pax1*	13	452	46,804.7	10.61	Nucleus
2	B. taurus_*Pax2*	26	469	50,169	8.82	Nucleus
3	B. taurus_*Pax3*	2	484	53,417.6	8.77	Nucleus
4	B. taurus_*Pax4*	4	364	39,307.9	9.38	Nucleus
5	B. taurus_*Pax5*	8	328	35,573.8	9.26	Nucleus
6	B. taurus_*Pax6*	15	422	46,653	9.7	Nucleus
7	B. taurus_*Pax7*	2	505	55,100.5	9.29	Nucleus
8	B. taurus_*Pax8*	11	457	48,762.8	7.93	Nucleus
9	B. taurus_*Pax9*	21	361	38,476.3	9.65	Nucleus

## Data Availability

No new data were created or analyzed in this study. Data sharing is not applicable to this article.

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
