# Peer review of "Genome-Wide Identification and Evolutionary and Mutational Analysis of the Bos taurus Pax Gene Family"

_genes, 2024, doi:10.3390/genes15070897_

Round 1
Reviewer 1 Report
Comments and Suggestions for Authors
1, Latin names are very differently presented - not italic, not capital letter when they should be
2. All studies are based on the data about 1 representant of every of 5 species. So there is no possibility to find intra species variability
3. The introduction is about Bos/Bubalis history so Sus scrofa or Ovis aries is not needed in the study
4. Very good using bioinformatics method, good quality presentation but the value of result is not large. There is no explanation about species relation in different pax genes. Also more interestng will be comparison to other species within Bovini
So in conclusion - good work but must be done on larger and another set of data
Author Response
Comment 1: Latin names are very differently presented - not italic, not capital letter when they should be
Response 1: Thanks for your suggestion. According to your guidance, we have changed the Latin names of all the animals in the article to italics.
Comment 2: All studies are based on the data about 1 representant of every of 5 species. So there is no possibility to find intra species variability
Response 2: Thanks for your suggestion. Most kinds of research will set multiple samples to ensure the accuracy of the experiment and explore the variability within the target species population. In the study of gene families, we usually only study each species using the officially published reference genome of the target species. Because the genome-wide data published by individuals often lack sufficient complete and accurate annotation files, it is difficult to extract the sequence of the gene family members of the current species from them. Since gene family analysis started from the research on plants, and the research on animals started relatively late, most of the literatures listed below are those on plants, but the basic principles and research methods are the same.
- Hussain Q, Ye T, Li S, Nkoh JN, Zhou Q, Shang C. Genome-Wide Identification and Expression Analysis of the Copper Transporter (COPT/Ctr) Gene Family in Kandelia obovata, a Typical Mangrove Plant. Int J Mol Sci. 2023 Oct 25;24(21):15579. doi: 10.3390/ijms242115579. PMID: 37958561; PMCID: PMC10648262.
- Yan J, Su P, Meng X, Liu P. Phylogeny of the plant receptor-like kinase (RLK) gene family and expression analysis of wheat RLK genes in response to biotic and abiotic stresses. BMC Genomics. 2023 May 1;24(1):224. doi: 10.1186/s12864-023-09303-7. PMID: 37127571; PMCID: PMC10152718.
- Sun W, Ma Z, Chen H, Liu M. MYB Gene Family in Potato (Solanum tuberosum L.): Genome-Wide Identification of Hormone-Responsive Reveals Their Potential Functions in Growth and Development. Int J Mol Sci. 2019 Sep 29;20(19):4847. doi: 10.3390/ijms20194847. PMID: 31569557; PMCID: PMC6801432.
- Niimura Y, Tsunoda M, Kato S, Murata K, Yanagawa T, Suzuki S, Touhara K. Origin and Evolution of the Gene Family of Proteinaceous Pheromones, the Exocrine Gland-Secreting Peptides, in Rodents. Mol Biol Evol. 2021 Jan 23;38(2):634-649. doi: 10.1093/molbev/msaa220. PMID: 32961551; PMCID: PMC7826187.
Comment 3: The introduction is about Bos/Bubalis history so Sus scrofa or Ovis aries is not needed in the study
Response 3: Thanks for your comment. We only added data from Sus scrofa and Ovis aries at the beginning of the evolutionary tree. Bos taurus, Bos indicus, Bubalus bubalis, Bos Sus scrofa and Ovis aries are common domestic animals that were domesticated by humans long ago and all have members of the pax gene family in their genomes. Adding them to the evolutionary tree at step 1 will help us better see the tight relationships among the members of the gene family in Bos taurus, Bos indicus and Bubalus bubalis, some of the gene family members of Bos taurus, Bos indicus, and Bubalus bubalis also showed significant differentiation. Such preliminary knowledge can help us to add more evidence for the subsequent evaluation of the relationship between the function of pax gene family members in Bos taurus, Bos indicus and Bubalus bubalis, and can help us to better predict the function of pax gene family members in Bos taurus.
Comment 4: Very good using bioinformatics method, good quality presentation but the value of result is not large. There is no explanation about species relation in different pax genes. Also more interestng will be comparison to other species within Bovini
Response 4: Thanks for your comment on the analysis process and discussion summary. We have redone part of the analysis to reflect the relationship between pax gene family members in Bos_taurus, Bos indicus, and Bubalus bubalis (lines 190 -216). This makes the overall experimental process smoother and the results clearer. We also optimized the discussion and conclusion sections (lines 249-346).
Reviewer 2 Report
Comments and Suggestions for Authors
Brief summary. The Pax gene family is crucial in mammals, contributing to the development of the nervous system and various tissues and organs. Earlier research on the Pax gene family did not focus on cross-species comparisons or the evolutionary links between gene families in different species. This study uniquely examined the differences and evolutionary relationships of the PAX gene family between buffalo and domestic cattle, utilizing bioinformatics techniques to identify a new molecular target for enhancing meat yield performance in buffalo breeding.
General concept comments
1. Introduction. The introduction of the article contains an extensive review of the relevant literature related to the topic being examined.
2. Materials and Methods. The study design is well-documented and clearly described, providing a clear outline of how the research was conducted.
3. Data management and statistical evaluation. In the methodology section, I recommend separating the statistical analysis part, detailing the statistical analysis methods used, and specifying the software employed..
4. From my perspective, the assessment of the results presented in the article aligns with the research methodology.
In this study, nine Pax genes were identified in the Bos taurus genome, classified into four subfamilies: Pax1/9, Pax2/5/8, Pax3/7, and Pax4/6. These Pax gene family members were located in the nucleus. Molecular phylogenetic analysis indicated that the structure and sequence of Pax genes in buffalo and Bos taurus were similar. The distribution of species and quantity of domains and motifs within subfamily members was consistent. Furthermore, evolutionary analysis revealed that the Pax domain is highly conserved across all Pax gene family members. Protein interaction network analysis indicated that only the Pax3 and Pax7 genes had direct interactions among the PAX genes.
5. The research conclusions are logical and consistent with the results.
Specific comments
- In my opinion, the authors of the article should formulate a research hypothesis and clearly define the objectives of the study in the introduction section.
- The research sample and the statistical analysis methods applied should be described in a separate methodology section.
- Figures 4 and 7 may be difficult for readers to read and understand.
This study identified 44 members of the Pax gene family across five species using bioinformatic techniques. Phylogenetic analysis of Pax genes from these species was conducted to explore differences in affinities, divergences, and motifs. This analysis provided an in-depth understanding of the evolution of Pax family genes. The results indicated that Bos taurus genes have diverged over the course of evolution, and that not all genes share the same relationships. The affinities of Pax genes varied within the Pax1/9, Pax2/5/8, Pax3/7, and Pax4/6 subfamilies among buffalo and domestic cattle.
In my opinion, the work done by the authors is relevant and innovative from a theoretical and practical point of view.
Sincerely, reviewer.
Author Response
Comment 1: In my opinion, the authors of the article should formulate a research hypothesis and clearly define the objectives of the study in the introduction section.
Response 1: Thanks for your suggestions. According to your guidance, we have added a specific description of the research purpose of this manuscript at the end of the Introduction chapter, please see lines 70-75.
Comment 2: The research sample and the statistical analysis methods applied should be described in a separate methodology section.
Response 2: Thanks for your suggestion. We have made a complete presentation of the research samples and the overall research process in the Materials and Methods part of the article. In lines 77-94 of section 2.1, we give a detailed description of the samples used in the study, and the application analysis schemes used in the study are also given a detailed description in chapters 2.2-2.7 of the document (lines 95 - 143).
Comment 3: Figures 4 and 7 may be difficult for readers to read and understand.
Response 3: Thanks for your advice. Figures 4 has been changed to Table 1, which will help readers understand our work better. Figures 7 has been changed to Figures 6, which will make the figure easier to read.
Round 2
Reviewer 1 Report
Comments and Suggestions for Authors
Very little small things. Paper was improved